# Evaluating frequency and quality of pathogen-specific T cells

Nadia Anikeeva[1], Dolores Grosso[2], Neal Flomenberg[2,3] & Yuri Sykulev[1,2,3]

It is generally accepted that enumeration and characterization of antigen-specific T cells provide essential information about potency of the immune response. Here, we report a new technique to determine the frequency and potency of antigen-specific CD8 T cells. The assay measures changes of intracellular $Ca^{2+}$ in real time by fluorescent microscopy in individual CD8 T cells responding to cognate peptides. The T cells form continuous monolayer, enabling the cells to present the peptides to each other. This approach allows us to evaluate the kinetics of intracellular $Ca^{2+}$ signalling that characterizes the quality of T cell response. We demonstrate the usefulness of the assay examining the frequency and quality of cytomegalovirus-specific CD8 T cells from healthy donor and patient after haploidentical stem cell transplantation. The new assay has a potential to provide essential information determining the status of the immune system, disease morbidity, potency of therapeutic intervention and vaccine efficacy.

[1] Department of Microbiology and Immunology, Thomas Jefferson University, Philadelphia, Pennsylvania 19107, USA. [2] Department of Medical Oncology, Thomas Jefferson University, Philadelphia, Pennsylvania 19107, USA. [3] The Sidney Kimmel Cancer Center, Thomas Jefferson University, Philadelphia, Pennsylvania 19107, USA. Correspondence and requests for materials should be addressed to N.F. (email: Neal.Flomenberg@jefferson.edu) or to Y.S. (email: Yuri.Sykulev@Jefferson.edu).

The frequency of pathogen-specific and tumour-specific T cells and their functional activity reflect the effectiveness of immune responses and can serve as useful diagnostic and prognostic indicators[1–3]. Increase in intracellular concentration of $Ca^{2+}$ during T-cell activation appears to be a versatile marker of responding T cells[4,5] that is determined by the specificity of responding T cells but does not depend on the stage of T-cell differentiation and the spectrum of produced cytokines. Estimated 75% of all activation-regulated genes showed dependence on $Ca^{2+}$ flux[6]. This emphasizes the role of $Ca^{2+}$ signalling in regulating early signalling events, which influence functional T-cell responses[7]. Typically, $Ca^{2+}$ response of T cells induced by antigen stimulation is evaluated by flow cytometry using intracellular $Ca^{2+}$ indicators. However, the frequency of a small number of antigen-specific T cells is difficult to detect by flow cytometry assay due to large differences in the fluorescent intensity between the individual cells within heterogeneous T-cell population[8]. To overcome this drawback, we developed an approach that measures the $Ca^{2+}$ response in individual T cells by means of fluorescent microscopy. Specifically, we utilized $CD8^{+}$ T cells labelled with $Ca^{2+}$-dependent fluorophore and analyzed intracellular fluorescence of these T cells in monolayers before and after stimulation with specific antigenic peptides. Subtraction of intracellular fluorescent intensity measured before and after the stimulation at various time points revealed responding T cells and the kinetics of intracellular $Ca^{2+}$ accumulation. Using T-cell clones, we optimized the assay parameters and determined the limit of detection and sensitivity of the approach. We have found that <0.1% of responding T cells that are capable of fluxing $Ca^{2+}$ in a population of CD8 T cells could be reliably detected. We also determined that up to 100 different peptides could be tested in one round of the assay, which is important for testing of peptide pools in clinical applications. To demonstrate the usefulness of the approach, we analyzed frequency of cytomegalovirus (CMV)-specific T cells derived from peripheral blood of healthy donor and patient who underwent haploidentical stem cell transplantation.

In conclusion, the proposed novel assay permits examining the frequency and quality of antigen-specific human CD8 T cells. The new assay revealed potential diagnostic and prognostic values. It can be utilized to derive essential information determining the status of the immune system, disease morbidity, potency of therapeutic intervention and vaccine efficacy.

## Results

### Principal of the assay.
All nucleated cells, including CD8 T cells that play essential role in virus- and tumour-specific immunity, express MHC-I proteins. Thus, the T cells could recognize cognate peptide-MHC (pMHC) ligands on their surface, get activated and exercise effector functions against each other[9]. We exploited this essential property to identify CD8 T cells with the specificity of interest in a population of CD8 T cells isolated from human peripheral blood. Antigenic peptides added to the cells rapidly bind to available MHC class I on the surface of the T cells assembled into the monolayer resulting in appearance of cognate pMHC recognizable by the T-cell receptor (TCR). The recognition of the cognate pMHC on T-cell surface by a neighbouring T cell leads to $Ca^{2+}$ flux in responding cells. $Ca^{2+}$ flux is detected by measuring increase of intracellular fluorescent intensity in responding T cells labelled with calcium-dependent fluorophore by means of fluorescent microscopy. The difference between initial intracellular fluorescence and the fluorescence determined after the peptide addition in individual cells with MetaMorph software reveals frequency of responding T cells in the monolayer. The analysis of kinetics of increase in intracellular

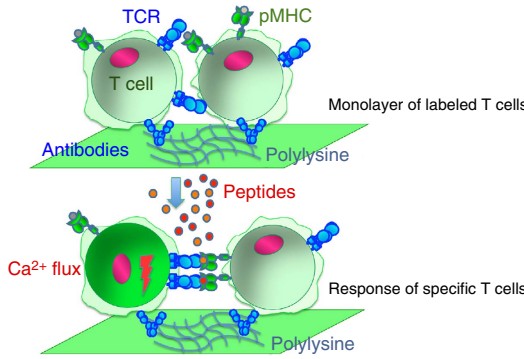

**Figure 1 | Schematic presentation of CaFlux assay.** T cells form monolayers on the glass bottom of the well covered with antibodies against T-cell's surface receptors. A peptide antigen is injected into the well. The peptide molecules bind to MHC proteins expressed on the T-cell surface to form recognizable pMHC ligands. Each T cells can serve as a target cells for neighbouring T cells, but only rare antigen-specific T cell is expected to recognize and respond to defined antigenic peptide used in the assay. Recognition events induce rise in the level of intracellular $Ca^{2+}$. To detect $Ca^{2+}$ influx, the T cells are labelled with $Ca^{2+}$ sensitive fluorophore. The changes in fluorescent intensity in individual cells before and after peptide addition are measured with fluorescent microscopy to identify the frequency of responding T cells.

$Ca^{2+}$ provides the information about efficiency of T-cell response to particular antigen. The assay is termed CaFlux assay and is schematically illustrated in Fig. 1.

### The assay development.
Glass bottom of 96-well plates was covered with poly-L-Lysine to capture TS2/4 antibody recognizing LFA-1 adhesion receptor without blocking LFA-1 functional activity. Cloned CD8 T cells with known specificity or polyclonal CD8 T cells were labelled with $Ca^{2+}$ fluorophore Fluo-4 and added to the wells. The quality of the T-cell monolayers was evident from analysis of bright field images of the immobilized T cells on the glass bottom of the plate. Figure 2a shows that T cells form uniform and tight monolayers allowing the T cells to contact each other, which is necessary for presentation and recognition of pMHC on one T cell by another T cell.

To optimize the conditions of the assay, we utilized established human clones of cytotoxic T lymphocytes (CTL) recognizing viral peptides in association with HLA-A2 MHC class I protein (see 'Methods' section). Addition of a cognate peptide to T-cell monolayer resulted in significant increase of intracellular fluorescent intensity of the T cells over the background indicative of $Ca^{2+}$ influx (Fig. 2b–d and Supplementary Movie 1). The $Ca^{2+}$ influx was observed in 90–95% of the T cells. The addition of non-cognate peptide to the monolayer did not induce intracellular $Ca^{2+}$ flux (Fig. 2d and Supplementary Movie 2). Because calcium ions distributed all over cytoplasm, bleaching of the fluorescence was not evident allowing to acquire dozens of images from the same field after the signalling was initiated (Supplementary Fig. 1).

The time course of the CD8 T-cell response to agonist peptide revealed two phases (Fig. 2d), which are usually observed in a population of responding T cells[10]. The first phase is characterized by a quick rise of intracellular $Ca^{2+}$ concentration that reaches maximum, followed by a decrease in the $Ca^{2+}$ level. Subsequently, the elevated level of intracellular $Ca^{2+}$ concentration is sustained over prolong period of time.

As opposed to functional T-cell responses such as production of cytokines or killing, $Ca^{2+}$ flux can be measured at room

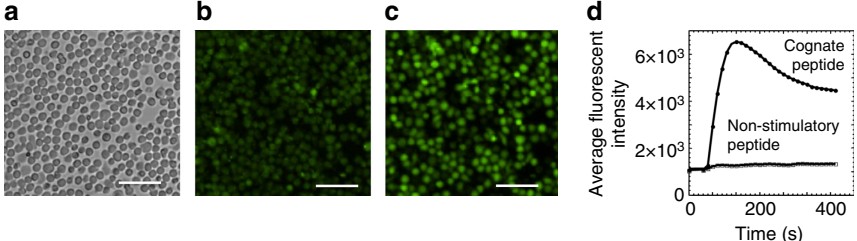

**Figure 2 | Detection of responding T cells in the monolayer.** Cloned CD8[+] T cells CER43 labelled with Fluo-4 were attached to a glass bottom of 96-well plates covered with poly-L-lysine via non-blocking anti-LFA-1 antibodies to form a monolayer ensuring direct contact of T cells with each other (**a,b**). Addition of a strong agonist peptide GILGFVFTL (GL9) to the monolayer of the labelled T cells induced intracellular $Ca^{2+}$ flux resulting in the time-dependent increase of intracellular fluorescence intensity that reaches maximum in 2–3 min (**c,d**). Addition of non-stimulatory peptide ILKEPVHGV (IV9) did not lead to the increased of intracellular fluorescence (**d**). Data are representative of nine independent experiments. Images (**b**) and (**c**) are taken at time 0 and 220 s after the stimulation, respectively. The results shown on **d** are based on the analysis of 4,500 T cells. Scale bars, 50 μm.

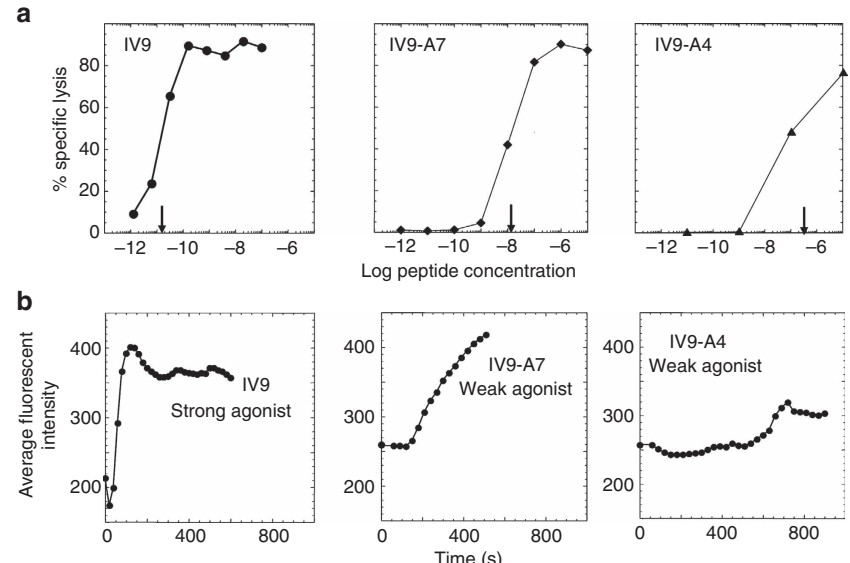

**Figure 3 | Strength of stimulation influences the kinetics of $Ca^{2+}$ signalling in CTL.** (**a**) Cytotoxic activity by human 68A62 cell line against HLA-A2[+] JY target cells is shown. Differences in concentration of strong ILKEPVHGV (IV9) and weak (IV9-A7 and IV9-A4) agonist peptides that are required to achieve the same extent of specific target cells lysis by 68A62 cells are marked by arrows. (**b**) Difference in the kinetics of calcium accumulation in responding 68A62 CTL stimulated with strong (IV9) and weak (IV9-A7 and IV9-A4) agonist peptides at a high peptide concentration (10$^{-4}$ M) is shown. The data are representative from two to five independent experiments. The results shown on **b** are based on the analysis of 2,000 T cells.

temperature simplifying the assay procedure and increasing the accuracy of the measurements. We compared $Ca^{2+}$ response at 37 °C, 32 °C and room temperature (22–24 °C). Although the kinetics of the response was slower at room temperature, lowering temperature did not significantly influence an optimal time window for measuring $Ca^{2+}$ signal and the ratio of maximal fluorescence intensity to the background fluorescence (Supplementary Fig. 1). Higher temperature increases leakage of the fluorophore from labelled cells and boost cell motility. Both factors significantly hinder image analysis, so all quantitative measurements were performed at room temperature.

To evaluate the capability of the assay to detect T cells whose TCR binds to its natural ligand with a low intrinsic affinity, we resorted to IV9-A7 and IV9-A4 variants of a strong agonist peptide ILKEPVHGV (IV9) recognizable by 68A62 CTL[11]. All three peptides bind equally well to HLA-A2, but have different potency in CTL assay. Compared with IV9, the concentration of IV9-A7 and IV9-A4 peptides required to achieve similar extent of specific target cell lysis was 3 and 4 orders of magnitude higher, respectively (Fig. 3a). The two weak agonist peptides were still

capable to elicit $Ca^{2+}$ flux in the monolayer of 68A62 CTL, but achieving the maximum of $Ca^{2+}$ increase in response to IV9-A7 stimulation was delayed to 10–15 min as compared with IV9 that induced maximum of $Ca^{2+}$ response within 2–3 min following stimulation (Fig. 3b). The response to another very weak agonist V9-A4 was barely detectable (Fig. 3b). Importantly, time required to achieve similar level of $Ca^{2+}$ increase induced by weak agonists was significantly prolonged (Fig. 3b). These results are consistent with previously published data showing that recognition of a strong agonist ligand results in a very rapid onset and fast accumulation of intracellular $Ca^{2+}$, while stimulation with weak agonists significantly delays the onset time and induces much slower accumulation of intracellular $Ca^{2+}$ with lower average magnitude[12,13]. Furthermore, the delay of the onset time revealed a poor dependence on the peptide concentration[12]. The data presented on Fig. 3b were produced at very high peptide concentration (10$^{-4}$ M), that is, conditions at which the difference in T-cell responses may be diminished. Nevertheless, the observed kinetics of $Ca^{2+}$ accumulation for strong and weak agonist ligands was clearly different. The difference in kinetics of

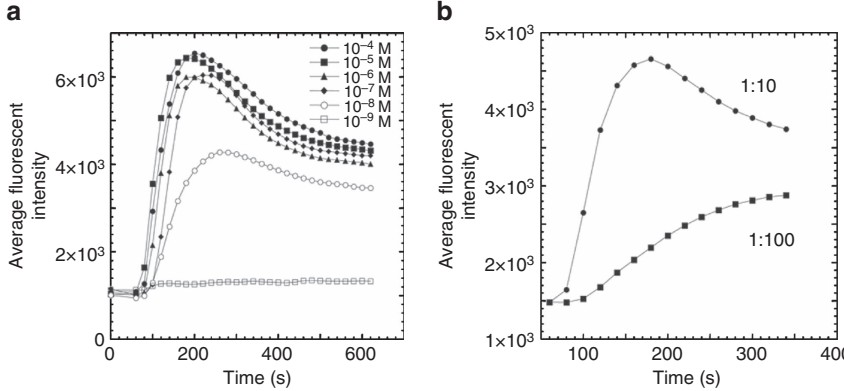

**Figure 4 | Breadth of tested peptides and the sensitivity of CaFlux assay.** (**a**) The dynamics and magnitude of calcium response by CD8$^+$ T cells CER43 stimulated by strong agonist peptide GILGFVFTL (GL9) at various concentrations. (**b**) The dependence of the sensitivity of calcium response in CD8 T cells CER43 induced by GL9 peptide ligand diluted in a mixture of non-stimulatory peptides (see 'Methods' section for details) at total concentration 10$^{-6}$ M and indicated ratios. The data are representative of 2 independent experiments and are based on the analysis of ∼5,000 cells.

calcium signalling induced by IV9, IV9-A7 and IV9-A4 peptides correlated with the killing potency induced by the same peptides. This is in accord with our previous findings showing that the magnitude and kinetics of Ca$^{2+}$ signalling control the efficiency CTL response[14,15].

To determine minimal peptide concentration that is required to detect specific T cells, we measured Ca$^{2+}$ response in CER43 CTL induced by a strong agonist peptide GL9 at various concentrations. Figure 4a shows the dependence of the calcium response by the T cells at peptide concentrations ranging from 10$^{-4}$ M to 10$^{-9}$ M. The time courses of Ca$^{2+}$ influx were similar at peptide concentration from 10$^{-4}$ M to 10$^{-7}$ M, and the peptide concentration of 10$^{-8}$ M appeared to be a minimal required concentration to induce detectable Ca$^{2+}$ response. This is about one order of magnitude higher than the sensitivity of specific lysis of GL9-sensitized target cells by CER43 CTL[16–18]. Comparison of 3D plot images illustrating Ca$^{2+}$ responses of the T cells at high (10$^{-4}$ M) and suboptimal (10$^{-8}$ M) peptide concentrations showed that the decrease of the response magnitude at lower peptide concentration was due to a lower amplitude of the responses of individual cells, but not due to the changes in the number of the responding cells (Supplementary Figs 2 and 3).

Virus infection commonly stimulates heterogeneous population of T cells with specificity to multiply virus epitope. We next tested the ability of CaFlux assay detecting T-cell responses to multiple peptide epitopes restricted by the same allele in one round of the assay. Each peptide in a mixture of several test peptides added to the monolayer of T cells isolated from peripheral blood could function as a cognate peptide for some T cells, while other peptides would behave as non-stimulatory or self-like peptides. Because T cells in the monolayer present peptide to each other, antigenic peptides restricted by different MHC alleles would function as self-like peptides. To this end, all T cells that are specific for each of the tested peptides are expected to respond to their respective cognate peptides providing that the concentration of these peptides is sufficient.

We chose Flu-derived peptide GL9 diluted in a mixture of non-stimulatory peptides (see 'Methods' section for details) at 1:10 and 1:100 ratios. We then determined whether the diluted GL9 was still capable to induce Ca$^{2+}$ flux in CER43 T cells. The number, which defines the fold excess of non-stimulatory peptides, is essentially equal to the number of stimulatory peptides that could be successfully tested in a single round of the assay. As evident from Fig. 4b, T-cell responses can be readily detected with a 10-fold excess and even 100-fold excess of non-stimulatory peptides suggesting that up to 100 peptides can still be tested simultaneously in one round of the assay. The latter is particularly important for clinical applications. To nail down which peptide(s) serve as T-cell epitopes for a sample of polyclonal T cells, multiple rounds of the assay with a smaller peptide groups and individual peptides have to be carried out.

To evaluate the sensitivity of the assay, we combined CER43 CD8 T cells and irrelevant 115iX HLA-A2$^+$ CTL at various ratios and determined the minimal number of responsive T cells that could still be detected. In these experiments, the CER43 T cells were utilized as antigen-specific T cells, while 115iX HLA-A2$^+$ CTL clone utilized as unresponsive T cells. The monolayers containing only 115iX CTL were served as a negative control. We usually observed a few false positive cells per imaging field containing 4,000–10,000 cells. The false positive signals were detected occasionally due to movement of non-adhered cells after peptide injection or accidental cell burst or transient rise of intracellular Ca$^{2+}$ concentration. The number of false positive cells slightly increased with time mostly due to the movements of non-adherent cells. To minimize the assay error, we calculated the frequency of T cells responding to strong agonist peptide at 3–4 min after injection. The data presented in Supplementary Table 1 show that as little as 3–5 responding T cells among ∼10$^4$ CER43 CTL per imaging field were reliably detected. In experiments with another CTL clone 68A62, we determined that <0.1% of antigen-specific T cells could be reliably identified (Supplementary Fig. 4).

**Application of the assay**. To demonstrate usefulness of CaFlux assay in measuring the frequency of pathogen-specific T cells, we utilized frozen samples of commercially available human peripheral blood mononuclear cell (PBMC) from healthy donors with known frequency of T cells specific for cytomegalovirus-derived peptide NLVPMVATV (NV9). The frequency of the NV9-specific T cells producing INF-γ was determined in ELISpot assay by the company. Supplementary Table 2 shows that the frequency of the CD8$^+$ T cells responding to either individual agonist peptide NLVPMVATV or ProMix HCMV specific peptide pool (Supplementary Table 3) was very similar despite a significant difference in concentration of the individual peptide alone (10$^{-4}$ M) or within the peptide pool (6 × 10$^{-6}$ M) during stimulation (Supplementary Fig. 5). We also determined the frequency of responding T cells in this sample by ELISpot

assay which was very similar to that measured by the CTL company (Supplementary Table 2). However, the frequency of CMV-specific T cells measured by ELISpot assay was about two times lower (Supplementary Table 2). The most plausible reason for the observed difference is that ELISpot assay counted only INF-γ producing cells, while the CaFlux assay detects all responding cells capable to flux $Ca^{2+}$ and mount various functional responses. In addition, terminally differentiated and exhausted cells could die during lengthy incubation (24–48 h) before producing enough cytokines to be measured by ELISpot assay. Results of the tetramer staining revealed a larger frequency of T cells specific for NV9-HLA-A2 ligand as compared with CaFlux (Supplementary Fig. 6 and Supplementary Table 2). The observed difference could be due to presence of unresponsive cells such as apoptotic cells, terminally differentiated and exhausted cells, and also the cells with low level of TCR and/or co-receptor.

We also examined the frequency of CMV-specific $CD8^+$ T cells in frozen samples of PBMC derived from a patient who has undergone bone marrow transplantation. In this patient, CMV reactivated on day 22 after the transplantation. The PBMC sample was taken on day 152 followed transplantation. For the induction of the response we utilized ProMix HCMV specific peptide pools from Thinkpeptides (Supplementary Table 3). The pool consists of 14 peptides representing the key immunodominant epitopes of human CMV covering 9 of the most relevant HLA types in human population. We found that the number of $CD8^+$ T cells was 25% of initially derived patient's PBMC as compared with 10% in healthy donor PBMC. The percentage of patient CMV-specific $CD8^+$ T cells was found to be $4.98 \pm 1.42\%$ (mean ± s.d., $n = 2$) or 13,825 responding CD8 T cells per $10^6$ PBMC. This is a very high frequency of the responding cells that were stimulated by two peptides presented by one MHC-I allele (HLA-A*01:01) (Supplementary Table 3). Thus, both parameters were significantly elevated as compared with those in samples of normal donors.

Comparison of the kinetics of CMV-specific CD8 T-cell responses in healthy donors and transplant patients revealed much faster response of the T cells from the donor as opposed to that from the transplant patient (Fig. 5 and Supplementary Fig. 7). Distinct response kinetics of healthy donor's and the patient's T cells is also evident from time-dependent appearance of responding T cells (Supplementary Movies 3 and 4). The kinetics of $Ca^{2+}$ signalling triggered by individual agonist peptide NLVPMVATV or ProMix HCMV specific peptide pool was found to be very similar (Supplementary Fig. 5 and Supplementary Movie 5). The response of the T cells from the patient recapitulate the response of $CD8^+$ T-cell clone towards weak agonist peptide (see Fig. 3a) suggesting that the T cells specific for the tested peptides appeared to be inefficient.

## Discussion

The CaFlux assay described here detects all T cells that are capable of responding to productive TCR ligation. Importantly, measuring the kinetics of $Ca^{2+}$ flux provides essential information regarding the efficiency of T-cell response, a major factor that characterizes the quality of the immune response. In contrast to CaFlux assay, ELISpot assay detects only those T cells that produce particular cytokines upon TCR stimulation. Another major difference between CaFlux and ELISpot assays is the time required for completion of the analysis, that is, few minutes versus 24–48 h, respectively. Not only does the difference in time required for measurements matter, but also the incubation of T cells for 24–48 h in the presence of stimulatory peptides could result in activation of T cells that initially were unresponsive.

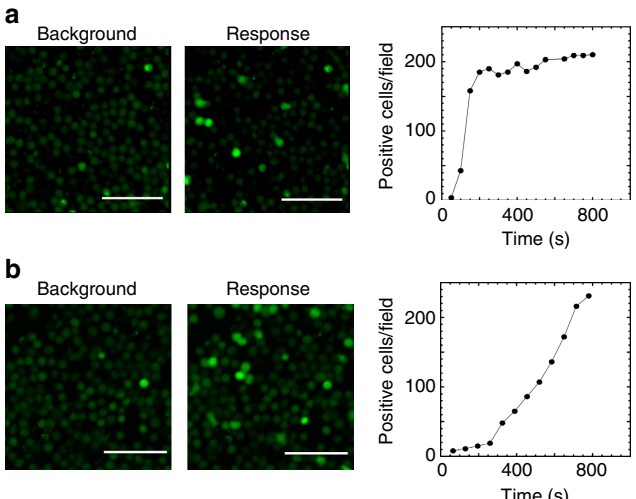

**Figure 5 | Kinetics of calcium response of CMV-specific CD8+ T cells.** CMV-specific CD8+ T cells from a healthy donor (**a**) and a patient after haploidentical stem cell transplantation (**b**) revealed different kinetics of $Ca^{2+}$ signalling on antigen stimulation. CD8+ T cells were purified from PBMC by negative magnetic sorting and purified CD8 T cells were labelled with Fluo-4 and immobilized on a glass surface as described in 'Methods' section. The T cells from healthy donor were stimulated by CMV-derived peptide NLVPMATV at $10^{-4}$ M while the T cells from the patient were triggered with ProMix CMV peptide pool at $9 \times 10^{-5}$ M. The images of responding cells are taken at 720 s after the stimulation. The data shown are representative of 5 (**a**) and 2 (**b**) independent experiments and are based on the analysis of 4,600 and 4,200 T cells, correspondingly. Scale bars, 50 μm.

Tetramer[19] and dextramer[20] binding assays typically detect only T cells whose TCR recognizes a given pMHC ligands but not necessarily respond to the ligand. Comparison of CaFlux with ELISpot and tetramer and dextramer detection of antigen-specific T cells is presented in Table 1.

It has to be noted that the assay design assumes that only $Ca^{2+}$ dependent T-cell responses could be detected. Anergic and exhausted and apoptotic cells could not flux calcium, and Fas/FasL-mediated T-cell response that may not depends on $Ca^{2+}$ flux[21–23]. These could preclude detection of tumour-specific T cells freshly isolated from solid tumours[24]. However, our goal is to detect responses that are linked to the ability of T cells to exercise functional pathogen and tumour-specific responses.

We have demonstrated the utility of CaFlux assay for measuring the frequency of T cells specific for a set of peptides. For instance, a set of CMV-peptide epitopes recognizable by protective T cells are already available, and measuring the frequency of T cells specific these peptide epitopes is an important diagnostic and prognostic parameter. Indeed, we have evaluated responses of CMV-specific T cells from a normal donor and a patient after bone marrow transplantation demonstrating the ability of the assay to distinguish between T-cell responses of different quality (Fig. 5). The frequency and quality of pathogen-specific T cells provide useful information as diagnostic, prognostic and safety biomarkers, and may serve as a predictor of clinical outcomes during immunotherapeutic interventions. Reliable immune monitoring of T-cell responses is also essential for vaccine development and adoptive T-cell transfer therapy as well as for analyzing responses against various T-cell epitopes in normal people to characterize the status of their immune system.

Thus far, we utilized the total fraction of $CD8^+$ T cells. But it is becoming evident that various subsets of CD8 T cells exercise

**Table 1 | Comparison of CaFlux with Tetramer and ELISpot assays.**

| Assay | CaFlux | Tetramer | ELISpot |
|---|---|---|---|
| What is measured | $Ca^{2+}$ signalling, a universal parameter of responding T cells | TCR specificity | Secreted cytokines |
| Time of the parameter measuring | 2–3 min (fluorescent microscopy) | 2–3 min (Flow cytometry) | 24–48 h plus measuring the level of secreted cytokines in culture supernatant |
| Type of detected CD8 T cells | Functional T cells plus the ability to evaluate the efficiency of T-cell response | Functional and unresponsive T cells | T cells producing defined cytokine and unresponsive T cells that could be activated during assay accounting for pseudo positive results |
| Frequency | A few cells per $10^4$ CD8 T cells | A few cells per $10^4$ CD8 T cells | A few cells per $2–4 \times 10^5$ PBMC* |
| Antigen-presenting cells | Not required but can be used | Not required | Required |
| Number of antigen-specific peptides per assay | 1–100 peptides restricted by a single or multiple MHC alleles | Up to 15 peptides | 10–20 peptides |
| Drawbacks | Required CD8 T-cell isolation | Requires production of pMHC protein for each peptide and detects unresponsive T cells | Detect CD8 T cells producing distinct cytokines and may detect initially unresponsive T cells |

*Because the fraction of CD8 T cells in PBMC correspond to 5–10%, the sensitivity of the assay is comparable to pMHC tetramer and CaFlux assays.

distinct functions[25–27] and could have different effect on clinical outcome. To analyze the subsets of the responding cells in CaFlux assay, magnetic sorting could be utilized to isolate T-cell subsets and their responses could then be evaluated. Alternatively, the assay would allow not only measuring $Ca^{2+}$ flux in the T cells, but also staining the cells with fluorescent-labelled antibodies for cell surface markers to visualize T cells of various phenotypes. The assay also has potential analyzing responses against various T-cell epitopes in normal people to characterize the status of their immune system.

This study provides a foundation for further development and clinical application of CaFlux assay. In the future, CaFlux assay could be adapted to characterizing T-cell responses in a course of autoimmune diseases including measuring the frequency of pathogenic CD4+ T cells.

## Methods

**T cells.** HIV- and Flu-specific human CD8+ T-cell clones, termed 68A62 and CER43, were kindly provided by Bruce Walker and Antonio Lanzavecchia, correspondingly. These T cells recognize ILKEPVHGV (IV9) and GILGFVFTL (GL9) peptides, respectively, both presented by HLA-A2 MHC class I (refs 14,17,18). 115iX is a CD8+ T-cell line developed from D3 CTL[28–30] as a result of spontaneous mutation in its TCR β chain resulting in loss of specificity for its natural ligand[31]. This cell line was used as T cells with irrelevant specificity for negative control. CTL were stimulated with a mixture of allogeneic irradiated PBMC and anti-CD3 antibody overnight. IL2 (50–200 units ml$^{-1}$) was then added to the culture, and the CTLs were allowed to rest for 12–17 days before the experiments. Human PBMC from healthy donors with known frequency of T cells specific for CMV-derived peptides were supplied by CTL Inc.

**Antibodies.** Hybridoma producing TS2/4 anti-LFA-1 antibodies was purchased from ATCC. The antibody was purified from culture supernatant by affinity chromatography on protein A Sepharose as described previously[18].

**Peptides.** Tax (LLFGYPVYV) peptide from human T-lymphotropic virus type 1 (ref. 32), SL9 (SLYNTVATL) peptide from HIV Gag p24 protein[28,29], MART-1 (ELAGIGILTV) peptide from Melan/MART-1 tumour antigen expressed on malignant melanoma cells[33] and GILGFVFTL (GL9) peptide from the influenza matrix protein[34] were synthesized by Research Genetics, Inc. ILKEPVHGV (IV9) peptide from HIV reverse transcriptase[11] and the peptide variants (IV9-A4 and IV9-A7) was a kind gift from Herman Eisen. NLVPMVATV (NV9) peptide from HCMV pp65 was purchased from CPC Scientific, Inc. ProMix HCMV peptide pool containing 14 different HCMV-derived peptides restricted by various MHC class I alleles was supplied by Thinkpeptides. The pooled HLA-A*0201 and

HLA-A*0101 restricted peptides are immunodominant peptides. These peptides bind to respected HLA proteins with high affinity[35,36] (also see Large scale analysis of peptide-HLA-I stability: http://www.iedb.org/refId/1028282).

To prepare non-stimulatory peptide mixture, Tax, SL9, MART-1 and IV9 peptides were combines at equimolar ratio and were utilized as non-cognate peptides, which did not induce antigen-specific $Ca^{2+}$ flux in CER43 cells.

*Magnetic sorting of T-cell subsets.* CD8 T cells were purified from frozen human PBMC by negative selection using MACS Cell Separation Technology according to manufacturer instruction (Miltenyi Biotec).

*T-cell labelling with calcium indicator.* A total of $10^6$ cells in 1 ml of PBS were loaded with Fluo-4 (Life Technologies) at 2–4 µM for 30 min at 37 °C in the presence of 0.02% pluoronic acid F-127 and 4 mM probenecid. The cells were washed free of unreacted reagents and incubated at 37 °C for additional 30 min. The cells were then resuspended in the assay buffer (20 mM HEPES, 140 mM NaCl, 5 mM KCl, 0.7 mM $Na_2HPO_4$, 6 mM D-glucose, 1 mM $CaCl_2$, 2 mM $MgCl_2$ and 1% HSA) and used for $Ca^{2+}$ flux analysis[14,37].

*Preparation of T-cell monolayers.* Glass bottom of 96-well MatTec plates was treated with 1 M HCL, thoroughly washed with water and covered with poly-L-lysine (Sigma, molecular weight > 300,000) at 0.1 mg ml$^{-1}$ for 1 h at room temperature. After washing with Dulbecco's PBS (DPBS), TS2/4 non-blocking mAb specific for LFA-1 were added to the plate at concentration 10 µg ml$^{-1}$ overnight & 4 °C. The wells were washed with DPBS, and $3 \times 10^5$ Fluo-4-labelled T cells in 100 µl of the assay buffer were added to each well. The plates were centrifuged at 100 g for one minute and were incubated for 30 min at room temperature before the imaging. The quality of T-cell monolayer was assessed using bright field microscopy.

*Measurements of $Ca^{2+}$ flux in responding T cells.* To identify responding T cells we imaged T-cell monolayers before (background measurement) and after (response) addition of stimulatory peptides. The images of T-cell monolayers were taken at experimentally determined exposure times using Zeiss Axiovert 200 M inverted microscope equipped with temperature controlled system and Andor Zyla sCMOS camera and ×10 objectives. For CTL clones, the average intensity of images before and after the T-cell stimulation at various time points were determined by MetaMorph software. The frequency of responding T cells in polyclonal populations was also determined by MetaMorph software. First, images acquired after the stimulation with peptides were subtracted from the background images. Using the resulting image, the threshold higher than the basal oscillations was established. Second, background images acquired before T-cell stimulation were subtracted from images acquired after the stimulation with peptides of interest to derive images containing responding cells. The latter were subjected to Integrated Morphometry Analysis to examine a combination of morphometric parameters such as object size, fluorescent intensity and/or shape to allow singling out responding T cells. Time-dependence of average fluorescence intensity in the responding T cells revealed TCR-mediated kinetics of $Ca^{2+}$ response. The number of labelled cells per field was counted using Count Nuclei function of the MetaMorph Premium software.

*Cytolytic assay.* Lymphoblastoid target cells JY ($5 \times 10^3$) were washed, $^{51}$Cr-labelled and then combined with various amounts of stimulatory peptide of interest in 150 µl R10 (RPMI-1640 containing 10% fetal calf serum). 68A62 or CER43 or 115iX in 50 µl in of R10 were then added to final assay volume of 200 µl. The assay was

performed in 96-well round-bottom plates at an effector-to-target ratio of 5:1. The plates were incubated for four hours in a $CO_2$ incubator at 37 °C and $^{51}$Cr release was measured in 100 μl of supernatant from each well. Per cent specific lysis was determined as previously described[18,37,38].

*IFN-γ ELISPOT assay.* ELISPOT assay was performed according to the instruction from manufacturer who supplied ELISPOT assay kit (CTL. Inc.). ELISPOT 96-well plates were coated with capture antibody against IFN-γ overnight at 4 °C. Freshly thawed PBMC were placed at quadruplicates in the plate at a density $2 \times 10^5$ per well along with an antigenic peptide at final concentration $10^{-6}$ M. The stimulation was performed for 24-hours at 37 °C in $CO_2$ incubator. Plates were washed and developed using detection antibody against IFN-γ, Streptividin-AP and the substrate solution. The numbers of spots per well were determined using CTL ImmunoSpot analyzer and CTL ImmunoSpot Software. The final data were reported after subtracting the average response in unstimulated samples.

*Quantification of antigen-specific T cells.* The fraction of the responding cells was determined as ratio of the T cells that respond on antigenic stimulation to total number of antigen-specific T cells taking into the experiment. Fluo-4-labelled CER43 cells were mixed with unlabelled cells at ratio 1:10. The cells were plated on 96-well plate to form monolayer. To induce $Ca^{2+}$ flux GL9 cognate peptide were added at concentration $10^{-4}$ M. The bright fields and fluorescent fields were acquired before and after addition of the peptide. Each imaging field contained from 50 to 400 labelled cells. The pool of responding cell was found by subtraction of a background image from the image acquired after peptide addition. The subtracted image was compared with the background image to identify non-responding cells. The fraction of responding cells was calculated from three independent experiments and is equal to $92.8 \pm 3.9\%$ (mean ± s.d.).

**Data availability.** All data in this manuscript are available from the authors on request.

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

## Acknowledgements

This work was supported by Pharma Germinator program of Southwestern PA. We would like to acknowledge excellent technical assistance of Flow Cytometry and Bioimaging Facilites that were supported by 5P30CA056036-17 core grant to The Sidney Kimmel Cancer Center. We are grateful to Bruce Walker and Antonio Lanzavecchia for providing human CD8+ T-cell clones. We thank Maria Steblyanko for help with movie preparation and Shaul Kushinsky for performing some experiments characterizing CMV-responding CD8 T cells derived from human blood during his rotation in the laboratory. We also thank Christine McCauley and Karen Duffy (Janssen Pharmaceutical, Inc.) for helpful comments during discussion of the experimental data.

## Author contributions

N.A., Y.S., D.G. and N.F. conceived the project. N.A. and Y.S. designed and executed the experiments, D.G. and N.F. provided patient's blood sample and described the patient's clinical status. N.A., Y.S., D.G. and N.F. interpreted the data and wrote the manuscript.

## Additional information

**Competing financial interests:** The assay is protected by pending patent (US62/183,997), inventors Y.S., N.A., N.F. and D.G. The remaining authors declare no competing financial interests.

