## [Peer Review File · Nature Communications]

Reviewers' comments:

Reviewer #1 (Remarks to the Author):

The manuscript by Nadia Anikeeva et al. describes an assay that allows analysis of the frequency and responsiveness of antigen-specific CD8+ T cells. There is certainly and acute and growing need for new high throughput assay to enumerate antigen-specific T cells (and validate new T cell epitopes); the described CaFlux assay is original, straight forward and has high throughput capability. There are, however, several issues that should be considered when revising the manuscript.

1. The statement that the assay provides information on the functional activity of antigen-specific CD8+ T cells is questionable, because the assay measures initial intracellular Ca²⁺ elevations and not subsequent functional CD8+ T cell responses, which may not be properly gauged by initial Ca²⁺ flux (e.g. FasL up-regulation is Ca²⁺ independent, whereas TNF α and IFN γ responses are not, but their ratio can be differ). Moreover, the cytotoxicity data shown in Fig. 3 correlate poorly with the corresponding Ca²⁺ responses. What is the Ca²⁺ responses peptide concentration dependence?

2. In the Abstract it is stated that the described assay allows parallel detection of many peptides and in Table 1 up to 100 peptides are indicated. Because binding of peptides to MHC molecule can be of different strength, this statement is questioned. Moreover, because in this assay parallel testing of peptides precludes assessments of T cell frequencies for given specificities, its multiplexing capability seems limited.

3. The accuracy of the assay must be scrutinized and discussed more thoroughly. On page 5 it is indicated that 90-95 of cloned CTL respond, i.e. 5-10% of the cells are not detected. Why not? If one compares Figs. 2A-C remarkably large cell-to-cell variations of Ca²⁺ signals between unstimulated and stimulated cells are apparent. This issue needs to be addressed, namely because the authors claim that this assay has a detection limit of 0.03-0.05% specific cells and allows testing of up to 100 peptides in parallel. It would useful to add 3D plots showing fluorescence intensities of individual cells before and after stimulation and conclusively discuss the error of the method and its dependence on variables like input cell number, peptide concentration, time- and temperature dependence.

4. Regarding Table 1: It would be instructive to add comparative experimental data on the CMV specific CTL using the CaFlux assay (Fig. 5; Table S2) and the tetramer staining and IFN γ ELISPOT, i.e. widely used assays.

5. In chronic viral infections and cancer T cells can become exhausted/angenic T and exhibit aberrant or absent functional responses and tetramer staining. Does the described assay detect such T cells?

Minor Comments

1. The legends to the figures should be revised such that the reader understands their content without searching in the manuscript. The cells, peptides and experimental conditions need to be specified. The axes in figures should be properly labeled to clearly indicate what parameters are indicated.

2. On page 7 and in Table S1 it is indicated that 3-5 per 10,000 cells can be detected and also on page 7 that up to 100 peptides can be tested in parallel. More specific information and discussions should be provided on the number of cells required for these analyses, how many cells can/need to

be recorded and how the detection limits depends on these variables for given peptides and MHC restriction?

3. Regarding Fig. 5: i) what is the peptide concentration used and how does it impact the kinetic? How looks the kinetics, if one plots the average fluorescence intensities?

4. Liadi I et al. (J. Vis. Exp, 2013) reported a high throughput CTL assay based on fluorescence microscopy, which should be mentioned.

5. In the legend to Fig. 3B the peptide concentrations should be indicated: Moreover using equal x and y-axis scales would allow for better comparison between the panels.

Reviewer #2 (Remarks to the Author):

In the present manuscript N. Anikeeva et al. describe a method to estimate the frequency of antigen specific human CD8+ T lymphocytes within polyclonal T cell populations. They propose to use $[Ca^{2+}]_i$ increase (as detected by time-lapse microscopy) to identify individual antigen specific T cells.

This manuscript is interesting and is based on a clever idea: to exploit the capacity of human CD8+ T cells to present each other MHC Class I restricted peptides when immobilized on a glass surface to form a monolayer.

There are, however, several problems that limit my enthusiasm for the present manuscript.

- Results presented are not fully convincing since movies depicting the time-kinetics of changes in $[Ca^{2+}]_i$ levels in response to different stimuli are missing. It is important that the authors show typical movies showing the frequency of the responding cells within the entire fields.

- It is also important to specify how thresholding was performed and to what extent cells treated with low antigen concentrations (or with weak agonists) are different from cells left untreated (untreated cells can exhibit basal $[Ca^{2+}]_i$ oscillations).

An additional limitation is that the number of cells present in each field was estimated to be ~10.000 cells. The number of cells per field can vary from one experiment to another. An automated method to count the positive and negative cells in the fields during the time-lapse acquisition would make quantification more accurate.

- The authors write in the abstract that the proposed assay allows testing many different peptides recognizable by CD8+ T cells in a single round of analysis. However, to my opinion this statement is not true. In Figure 4, it is shown that the peptide of interest can be diluted within increasing concentrations of irrelevant peptides without losing the capacity to elicit $[Ca^{2+}]_i$ increase. However, if a polyclonal T cell population is treated with various peptides simultaneously it will be impossible to define to which peptide each individual cell is responding (since calcium fluxes will be similar in all responding cells).

- Images presented in Figure 5 A and B are not fully convincing, since it is not easy to appreciate differences in calcium levels between the two cell populations. Movies showing typical results would be of help.

Another question concerns how positive cells shown in right panels were scored. Were they scored using the Integrated Morphometric Analysis function of MetaMorph in the entire series of collected images? Results are representative of how many experiments?

- The figure legends should systematically report how many experiments were performed. For pooled calcium mobilization data the number of scored cells should be indicated.

Reviewer #3 (Remarks to the Author):

This solid assay development study by Anikeeva N et al described a straightforward and simple method (called CaFlux assay) to complement existing approaches to evaluate functions of antigen-specific CD8 T cells. The conclusions are strongly supported by reliable and compelling data.

CaFlux assay is based on real-time measurements of intracellular calcium by immobilizing CD8 T cells on a glass surface, followed by further addition of cognate peptides. The calcium flux assay has been very routinely used in many laboratories to assess TCR signaling, and thus the novelty of the study is unlikely to be high. In addition, comparable or similar methods for evaluating T cell functions (such as Tetramer and dextramers described in Refs 12-13) have been previously developed with superior sensitivity and efficiency. For future clinical use, the CaFlux assay is likely to be limited by the high demand on instruments and often, inadvertent preparation of CD8 T cells might pre-activate assayed samples. The reviewer did not see the obvious advantages of the CaFlux assay over other approaches.

Overall, this manuscript lacks novelty and represents incremental advance in assay development. Although it is a very solid and interesting manuscript, It might be most suitable for more specialized method development or immunology-related journals.

Some minor issues:

1. In the main text, the authors claimed "The time course of the CD8 T cell response to agonist peptide revealed two phases (Fig. 2D)". Nonetheless, only one phase of calcium influx was presented in panel D.
2. "Methods", first paragraph: it is recommended to present the unpublished data for characterizing the cell line used as a negative control.
3. "Methods" section, last paragraph: "...and IV9 peptides were combines at ...": need to be corrected as "were combined".
4. Please Justify why LFA-1 should be blocked in the assay for readers not in the immunology field.

Below are point-by-point answers to reviewers' questions.

Reviewer #1

The manuscript by Nadia Anikeeva et al. describes an assay that allows analysis of the frequency and responsiveness of antigen-specific CD8+ T cells. There is certainly and acute and growing need for new high throughput assay to enumerate antigen-specific T cells (and validate new T cell epitopes); the described CaFlux assay is original, straight forward and has high throughput capability. There are, however, several issues that should be considered when revising the manuscript.

1. The statement that the assay provides information on the functional activity of antigen-specific CD8+ T cells is questionable, because the assay measures initial intracellular Ca²⁺ elevations and not subsequent functional CD8+ T cell responses, which may not be properly gauged by initial Ca²⁺ flux (e.g. FasL up-regulation is Ca²⁺ independent, whereas TNF α and IFN γ responses are not, but their ratio can be differ). Moreover, the cytotoxicity data shown in Fig. 3 correlate poorly with the corresponding Ca²⁺ responses. What is the Ca²⁺ responses peptide concentration dependence?

Answer: The reviewer's comment is well taken. It is true that some CTL responses do not depend in Ca²⁺ signaling. However, 75% of all activation-regulated genes showed dependence on Ca²⁺ flux (**Nature Immunol, 2:316-324, 2001**). This emphasizes the role of Ca²⁺ signaling in regulating early signaling events, which influence functional CTL responses (**Immunology and Cell Biology, 93:694-704, 2015**). Moreover, efficiency of cytolytic activity, the major function of CTL, clearly depends on Ca²⁺ signaling. We have previously shown that the kinetics of Ca²⁺ mobilization, regardless of how strong Ca²⁺ signal might follow later, determines the kinetics of cytolytic vesicles delivery to the CTL/target cell interface, which is linked to the kinetics of target cell destruction (**Immunity, 31: 632-642, 2009; Sci Signal.,3: pe50, 2010; Immunol. Res., 51: 183-94, 2011**).

It has been shown that recognition of a strong agonist ligand results in a very rapid onset and fast accumulation of intracellular Ca²⁺, while stimulation with weak agonists significantly delays the onset time and induces much slower accumulation of intracellular Ca²⁺ with lower average magnitude (**J. Exp. Med., 185:1815-1825, 1997; J. Immunol.,184:1829-1839, 2010**). Furthermore, the delay of the onset time revealed a poor dependence on the peptide concentration (**J. Exp. Med., 185:1815-1825, 1997**). This is consistent with the data presented in **Figure 3**. Recognition of a strong agonist ligand, which induced sensitive and robust cytolytic response, resulted in a very rapid onset and fast accumulation of intracellular Ca²⁺, while weak agonists induced much slower accumulation of intracellular Ca²⁺ that is linked to a profound decrease of cytotoxic potency. The data presented on **Fig. 3** were produced at very high peptide concentration (10⁻⁴ M), i.e., conditions at which the difference in T-cell responses may be diminished. Nevertheless, the difference in the kinetics of Ca²⁺ accumulation for strong and weak agonist ligands is clearly evident. In addition, analysis of the 3D plot images at suboptimal peptide concentration (10⁻⁸ M) of cognate peptide (**Fig. 4 and Supplemental Figure 2**) revealed that the magnitude of the response reduced due to a lower amplitude of individual cell responses, but not due to the changes in the number of the responding cells. We clarified these issues in the revised manuscript.

2. In the Abstract it is stated that the described assay allows parallel detection of many peptides and in Table 1 up to 100 peptides are indicated. Because binding of peptides to MHC molecule can be of different strength, this statement is questioned. Moreover, because in this assay parallel testing of peptides precludes assessments of T cell frequencies for given specificities, its multiplexing capability seems limited.

Answer: The reviewer correctly noted that the specificity of T cells for 100 different peptides

couldn't be determined in one round of the assay. What we really meant is that 100 different peptides could be tested in one round of the assay, which is important for testing of peptide pools in clinical applications. Smaller groups of the peptides have to be analyzed to nail down which peptides serve as T cell epitopes for a sample of polyclonal T cells. We agree that peptides with significantly lower affinity may be missed. However, usually peptides that form sufficiently stable complexes with MHC proteins could induce potent T cell response and play essential role in the immune defense. The effect of the peptide competition on the stimulation of functional T cell response was studied with peptide pool comprising 32 well-characterized viral epitopes (**Viruses, 4: 2636-2649, 2012**). The effect of the competition was minor, - it causes around 20% decrease in the magnitude of the response. We have incorporated necessary comments into the revised manuscript to clarify this issue.

3. The accuracy of the assay must be scrutinized and discussed more thoroughly. On page 5 it is indicated that 90-95 of cloned CTL respond, i.e. 5-10% of the cells are not detected. Why not? If one compares Figs. 2A-C remarkably large cell-to-cell variations of Ca²⁺ signals between un-stimulated and stimulated cells are apparent. This issue needs to be addressed, namely because the authors claim that this assay has a detection limit of 0.03-0.05% specific cells and allows testing of up to 100 peptides in parallel. It would be useful to add 3D plots showing fluorescence intensities of individual cells before and after stimulation and conclusively discuss the error of the method and its dependence on variables like input cell number, peptide concentration, time- and temperature dependence.

Answer: Even though cloned T cells expressed the same TCR, they are not uniform. At the time when the T cells are tested after the stimulation, there are never 100% of the tested T cells capable to mount a response to cognate pMHC. For example, about 10% of unresponsive cells were observed even for Jurkat cells stimulated with surface-bound anti-CD3 antibodies (**PLOS One, 9: e85934, 2014**). There are various reasons for the unresponsiveness, namely, apoptosis, anergy, difference in the number of cell surface receptors etc. To measure CTL responsiveness in the CaFlux assay, we diluted the labeled cells with unlabeled CTL at ratio 1:10 and used them in the assay. This approach allows us to follow labeled cells and to quantify % of unresponsive cells. The major factor of the unresponsiveness is the cell conditions. Other factors include monolayer imperfections and heterogeneity of the cell labeling. We incorporated this information in the **Supplemental Methods** Section.

We agree with the reviewer that there are variations in the amount of fluorophore picked up by individual cells during labeling. This is a major obstacle for analysis of calcium response by Flow Cytometry as we have stated in the Introduction. In this assay, the fluorescence intensity of each individual cell is compared before and after stimulation that alleviated problems with heterogeneity of the labeling. The brightest cells are presumably apoptotic cells with leaky membrane. These cells are always present in very small amount and are removed from the analysis after image subtraction procedure.

We have prepared 3D plot showing fluorescence intensities of labeled T cells on the monolayer before and after stimulation with antigenic peptide at different peptide concentrations.

The figure is incorporated into **Figure S2** and **S3** of the manuscript. The 3D image provides excellent visualization of the assay and validates the results of the quantitative analysis of 2D images.

Input cell number: We typically analyzed at least three imaging fields per sample that represent totally about 15,000-30,000 CD8 T cells. This is similar to the amount of cells that is recommended to analyze in tetramer assay. We usually observed a few false positive cells per field. The false positive signals occur mostly due to movement of non-adhered cells after peptide injection or accidental cell burst or transient rise of intracellular Ca^{2+} . We also analyzed the assay linearity by diluting the antigen specific T cells and found that the linearity of the assay is preserved for tested CTL (**Figure S4**).

Peptide concentration: For strong agonist peptide, Ca^{2+} response has very similar characteristics within the peptide concentration range from 10^{-4} M to 10^{-7} M (**Fig. 4**). At the suboptimal peptide concentration the magnitude of response declined in each individual cell (**Figure S3**) that increases the assay error.

Time dependence: The number of false positive cells slightly increased with time mostly due to non-adherent cells. To minimize the method error, we calculate the frequency of the cells responding to strong agonist peptide at 3-4 min after the peptide injection, an optimal time

Temperature dependence: Although a response curve has similar shape at different temperatures, higher temperature leads to increased leak of fluorophore from labeled cells and enhanced cell motility. Both factors significantly hinder image analysis, so all quantitative measurements were done at room temperature.

4. Regarding Table 1: It would be instructive to add comparative experimental data on the CMV specific CTL using the CaFlux assay (Fig. 5; Table S2) and the tetramer staining and IFN γ ELISPOT, i.e. widely used assays.

Answer: We tested CMV-specific CTLs using ELISPOT and tetramer staining, and the results are incorporated into the revised manuscript (**Table S2, Fig. S6 and Supplemental methods**).

ELISPOT results for healthy donor: Background reading 15 ± 11 (mean \pm SD) per 200,000 PBMC; response to CMV specific NLVPMVATV peptide is 509.25 ± 17.46 (mean \pm SD) per 200,000 PBMC. The frequency of NLVPMVATV-HLA-A02 tetramer positive cells was found to be 10% among total CD8 $^+$ CD3 $^+$ T cells.

In agreement with previously published data (**Clinical and Developmental Immunology, article ID 451059, 2012**), the tetramer staining reveal larger amount of CMV-specific CD8 T cells in healthy donor as compared to ELISPOT assay. As we expected, the frequency of the CMV-specific T cells determined by CaFlux assay was higher compared to ELISPOT assay and lower than that found with tetramer staining. We incorporated these data into the revised manuscript.

5. In chronic viral infections and cancer T cells can become exhausted/angery T and exhibit aberrant or absent functional responses and tetramer staining. Does the described assay detect such T cells?

Answer: We agree and we expect that anergic and exhausted/apoptotic cells will not respond in the assay. However, our goal is to detect functional responses that are linked to the ability of T cells to exercise efficient virus- and cancer-specific responses. We propose that the efficient T-cell responses are expected to correlate with the ability of human immune system to fight viruses and cancer.

Minor Comments

1. *The legends to the figures should be revised such that the reader understands their content without searching in the manuscript. The cells, peptides and experimental conditions need to be specified. The axes in figures should be properly labeled to clearly indicate what parameters are indicated.*

Answer: We have revisited figure legends and made other necessary changes to make the figures more informative minimizing their dependence on the text.

2. *On page 7 and in Table S1 it is indicated that 3-5 per 10,000 cells can be detected and also on page 7 that up to 100 peptides can be tested in parallel. More specific information and discussions should be provided on the number of cells required for these analyses, how many cells can/need to be recorded and how the detection limits depends on these variables for given peptides and MHC restriction?*

Answer: This issue has been addressed above (see answer to major question 2).

3. *Regarding Fig. 5: i) what is the peptide concentration used and how does it impact the kinetic? How looks the kinetics, if one plots the average fluorescence intensities?*

Answer: We indicated peptide concentrations in **Fig. 5** Legend. **Fig. 5A:** 10^{-4} M of the peptide; **Fig 5B:** 8.75×10^{-5} M of the peptide mixture or 6×10^{-6} of each individual peptide. Since the pool contains NLVPMVATV peptide restricted by HLA-A2, we also utilized the peptide pool to induce the response in CD8 T cells from healthy donor. The frequency of the cells responding to the individual peptide and the peptide pool and the kinetics of the responses were found to be similar (**Fig. S5**).

We plotted the average fluorescence intensity upon time and found that the kinetics of the responses remained the same (**Fig. S7**). However the differences in the average fluorescent intensity over background were smaller due to relatively low frequency of the responding cells.

4. *Liadi I et al. (J. Vis. Exp, 2013) reported a high throughout CTL assay based on fluorescence microscopy, which should be mentioned.*

Answer: We overlooked this paper. We quote this work in the revised manuscript.

5. In the legend to Fig. 3B the peptide concentrations should be indicated: Moreover using equal x and y-axis scales would allow for better comparison between the panels.

Answer: The peptide concentrations have been indicated in the Figure Legend consistent with our answer to the Reviewer's minor comment #1. We equalized scales of x- and y-axes.

Reviewer #2

In the present manuscript N. Anikeeva et al. describe a method to estimate the frequency of antigen specific human CD8+ T lymphocytes within polyclonal T cell populations. They propose to use [Ca²⁺]_i increase (as detected by time-lapse microscopy) to identify individual antigen specific T cells. This manuscript is interesting and is based on a clever idea: to exploit the capacity of human CD8+ T cells to present each other MHC Class I restricted peptides when immobilized on a glass surface to form a monolayer.

There are, however, several problems that limit my enthusiasm for the present manuscript.

1. Results presented are not fully convincing since movies depicting the time-kinetics of changes in [Ca²⁺]_i levels in response to different stimuli are missing. It is important that the authors show typical movies showing the frequency of the responding cells within the entire fields.

Answer: We prepared the movie showing changes in fluorescent intensity of responding CTL vs time within the entire fields (**Movie 1**). We also prepared the movie showing CTL after treatment with a control peptide (**Movie 2**).

2. It is also important to specify how thresholding was performed and to what extend cells treated with low antigen concentrations (or with weak agonists) are different from cells left untreated (untreated cells can exhibit basal [Ca²⁺]_i oscillations). An additional limitation is that the number of cells present in each field was estimated to be ~10.000 cells. The number of cells per field can vary from one experiment to another. An automated method to count the positive and negative cells in the fields during the time-lapse acquisition would make quantification more accurate.

Answer: To perform thresholding, the image taken immediately after a peptide addition is subtracted from the background image. Using the resulting image, the threshold higher than basal oscillations is established. At suboptimal peptide concentration (10⁻⁸ M, **Fig. 4**) the response magnitude of each individual antigen-specific T cell is significantly higher than the basal oscillations (see 3D plot, **Fig. S3**). The above concentration (10⁻⁸ M) is close to suboptimal peptide concentration (10⁻⁹ M) of cytolytic response for given CTL clone. Very weak agonist peptide (**Fig. 3**, panel **B**, IV9-A4), which presumably unable to exercise physiologically important activity, induced significantly

diminished response with the oscillations reaching the baseline. Thus, the frequency of a low avidity T cells could be underestimated.

The number of labeled cells per field was counted using Count Nuclei function of the MetaMorph Premium software. This application module was created to segment images that are utilized to identify cell nuclei in order to count the cells. We found that the application works very well to identify labeled individual T cells and to determine the total number of labelled cells within a monolayer. We included this information into Methods Section and into the relevant Figure Legends.

3. The authors write in the abstract that the proposed assay allows testing many different peptides recognizable by CD8+ T cells in a single round of analysis. However, to my opinion this statement is not true. In Figure 4, it is shown that the peptide of interest can be diluted within increasing concentrations of irrelevant peptides without losing the capacity to elicit $[Ca^{2+}]_i$ increase. However, if a polyclonal T cell population is treated with various peptides simultaneously it will be impossible to define to which peptide each individual cell is responding (since calcium fluxes will be similar in all responding cells).

Answer: This is essentially the same question that was raised by reviewer #1 (see above question #2). What we really meant is that 100 different peptides could be tested in one round of the assay to determine whether some of them may induce T-cell response. This is important for testing of peptide pools in clinical applications. Smaller groups of the peptides then have to be tested to nail down which peptides serve as T cell epitopes for T cells within a given sample of polyclonal T cells.

4. Images presented in Figure 5 A and B are not fully convincing, since it is not easy to appreciate differences in calcium levels between the two cell populations. Movies showing typical results would be of help.

Another question concerns how positive cells shown in right panels were scored. Were they scored using the Integrated Morphometric Analysis function of MetaMorph in the entire series of collected images? Results are representative of how many experiments?

Answer: We prepared a movie showing changes in frequency of responding CD8 T cells from healthy donor and patient after bone marrow transplantation during time course of the data collection within the entire field (**Movie 3** and **Movie 4**). The difference in the kinetics of the response is clearly evident. The quick, robust and sustained response observed in case of healthy donor with only a few cells that have $[Ca^{2+}]_i$ oscillations reaching baseline level or transient response where Ca^{2+} signals returned to the baseline. The patient's T cells showed delayed in calcium response with $[Ca^{2+}]_i$ oscillation pattern.

The positive cells were scored in the entire series of collected images using the Integrated Morphometric Analysis function of MetaMorph. To verify the results, some images were also scored using Overlap Function of the MetaMorph. The background images were converted to red color

images, while the images with responding cells were converted to the green color images. After overlapping of the background and the responding images, the green responding cells were counted using Manually Count Object Function of MetaMorph.

The results shown are representative of 5 (healthy donor) and 2 (the patient) independent experiments.

5. The figure legends should systematically report how many experiments were performed. For pooled calcium mobilization data the number of scored cells should be indicated.

Answer: The number of experiments performed has been indicated in the revised Figure Legends. The number of cells analyzed in pooled data is also indicated.

Reviewer #3

This solid assay development study by Anikeeva N et al described a straightforward and simple method (called CaFlux assay) to complement existing approaches to evaluate functions of antigen-specific CD8 T cells. The conclusions are strongly supported by reliable and compelling data.

1. CaFlux assay is based on real-time measurements of intracellular calcium by immobilizing CD8 T cells on a glass surface, followed by further addition of cognate peptides. The calcium flux assay has been very routinely used in many laboratories to assess TCR signaling, and thus the novelty of the study is unlikely to be high. In addition, comparable or similar methods for evaluating T cell functions (such as Tetramer and dextramers described in Refs 12-13) have been previously developed with superior sensitivity and efficiency. For future clinical use, the CaFlux assay is likely to be limited by the high demand on instruments and often, inadvertent preparation of CD8 T cells might pre-activate assayed samples. The reviewer did not see the obvious advantages of the CaFlux assay over other approaches.

Overall, this manuscript lacks novelty and represents incremental advance in assay development. Although it is a very solid and interesting manuscript, It might be most suitable for more specialized method development or immunology-related journals.

Answer: We respectfully disagree with the reviewer that newly developed CaFlux assay lacks novelty. CaFlux assay has numerous advantages over existing assays that have been discussed in the Discussion of the manuscript. Specifically, the detection of Ca²⁺ in a small fraction of responding cells in polyclonal population of T cells by Flow Cytometry is not feasible. Meaningful measurements of the kinetics of Ca²⁺ mobilization with tetramer and dextramer is limited to cloned T cells. The preparation of freshly isolated human CD8 T cells by **negative** magnetic sorting for ex vivo analysis is now highly reproducible and routine procedure. Miltenyi Biotec Inc. developed a robot for purification of various populations of human T cells that is now used in hospitals. The quality of the purified T cells is suitable for

different kinds of analysis including CaFlux assay. Future automation of the assay is also feasible and CaFlux assay is expected to bring about another dimension to characterization of human T cells contributing to establishing criteria of the quality of human immune responses.

Some minor issues:

1. *In the main text, the authors claimed "The time course of the CD8 T cell response to agonist peptide revealed two phases (Fig. 2D)". Nonetheless, only one phase of calcium influx was presented in panel D.*

Answer: This description is consistent with previously observed changes in the level of intracellular Ca^{2+} followed by T cells stimulation with stimulatory pMHC ligands; namely, the calcium level initially increased and after reaching a maximum declines. The observation was done for total cell population and is usually detected by flow cytometry (**Cytometry, 73(3):246-253; 2008; PNAS, 103(45):16846-51, 2006**). We referred to these two processes to as two phases.

2. *"Methods", first paragraph: it is recommended to present the unpublished data for characterizing the cell line used as a negative control.*

Answer: We previously described this clone in J Clin. Invest. 113(1): 49-57, 2004.

3. *"Methods" section, last paragraph: "...and IV9 peptides were combines at ...": need to be corrected as "were combined".*

Answer: Appropriate correction has been made.

4. *Please Justify why LFA-1 should be blocked in the assay for readers not in the immunology field.*

Answer: Indeed, we have utilized non-blocking TS2/4 antibodies to LFA-1 that do not interfere with the LFA-1 engagement by its natural ligand ICAM-1; the latter mediates intercellular adhesion and induces LFA-1-mediated signaling that contributes to proximal signaling and influences T cell responses. This information has been provided in **The Assay Development** Section. We have also incorporated this information into **Fig. 2** legend.

REVIEWERS' COMMENTS:

Reviewer #1 (Remarks to the Author):

Answer 1: The reviewer's comment is well taken. It is true that some CTL responses do not depend on Ca^{2+} signaling. However, 75% of all activation-regulated genes showed dependence on Ca^{2+} flux (Nature Immunol, 2:316-324, 2001). This emphasizes the role of Ca^{2+} signaling in regulating early signaling events, which influence functional CTL responses (Immunology and Cell Biology, 93:694-704, 2015). Moreover, efficiency of cytolytic activity, the major function of CTL, clearly depends on Ca^{2+} signaling. We have previously shown that the kinetics of Ca^{2+} mobilization, regardless of how strong Ca^{2+} signal might follow later, determines the kinetics of cytolytic vesicles delivery to the CTL/target cell interface, which is linked to the kinetics of target cell destruction (Immunity, 31: 632-642, 2009; Sci Signal.,3: pe50, 2010; Immunol. Res., 51: 183-94, 2011). It has been shown that recognition of a strong agonist ligand results in a very rapid onset and fast accumulation of intracellular Ca^{2+} , while stimulation with weak agonists significantly delays the onset time and induces much slower accumulation of intracellular Ca^{2+} with lower average magnitude (J. Exp. Med., 185:1815-1825, 1997; J. Immunol.,184:1829-1839, 2010). Furthermore, the delay of the onset time revealed a poor dependence on the peptide concentration (J. Exp. Med., 185:1815-1825, 1997). This is consistent with the data presented in Figure 3. Recognition of a strong agonist ligand, which induced sensitive and robust cytolytic response, resulted in a very rapid onset and fast accumulation of intracellular Ca^{2+} , while weak agonists induced much slower accumulation of intracellular Ca^{2+} that is linked to a profound decrease of cytotoxic potency. The data presented on Fig. 3 were produced at very high peptide concentration (10^{-4} M), i.e., conditions at which the difference in T-cell responses may be diminished. Nevertheless, the difference in the kinetics of Ca^{2+} accumulation for strong and weak agonist ligands is clearly evident. In addition, analysis of the 3D plot images at suboptimal peptide concentration (10^{-8} M) of cognate peptide (Fig. 4 and Supplemental Figure 2) revealed that the magnitude of the response reduced due to a lower amplitude of individual cell responses, but not due to the changes in the number of the responding cells. We clarified these issues in the revised manuscript.

Comment: The revisions are fine. It should be mentioned that CTL have two cytotoxic mechanisms; one of which, the Fas/FasL pathway is not or very little Ca^{2+} dependent and hence is not detected by the described method (PMID: 24594598; PMID: 9522460; PMID: 9529322, PMID: 8666927). This cytotoxic pathway is important, e.g. for tumor control (PMID: 23341634).

Answer 2: The reviewer correctly noted that the specificity of T cells for 100 different peptides couldn't be determined in one round of the assay. What we really meant is that 100 different peptides could be tested in one round of the assay, which is important for testing of peptide pools in clinical applications. Smaller groups of the peptides have to be analyzed to nail down which peptides serve as T cell epitopes for a sample of polyclonal T cells. We agree that peptides with significantly lower affinity may be missed. However, usually peptides that form sufficiently stable complexes with MHC proteins could induce potent T cell response and play essential role in the immune defense. The effect of the peptide competition on the stimulation of functional T cell response was studied with peptide pool comprising 32 well-characterized viral epitopes (Viruses, 4: 2636-2649, 2012). The effect of the competition was minor, - it causes around 20% decrease in the magnitude of the response. We have incorporated necessary comments into the revised manuscript to clarify this issue.

Comment: The arguments and revisions are fine. It is good to make clear these issues, namely that multiple rounds are needed to identify peptide specificities.

Answer 3: Even though cloned T cells expressed the same TCR, they are not uniform. At the time

when the T cells are tested after the stimulation, there are never 100% of the tested T cells capable to mount a response to cognate pMHC. For example, about 10% of unresponsive cells were observed even for Jurkat cells stimulated with surface-bound anti-CD3 antibodies (PLOS One, 9: e85934, 2014). There are various reasons for the unresponsiveness, namely, apoptosis, anergy, difference in the number of cell surface receptors etc. To measure CTL responsiveness in the CaFlux assay, we diluted the labeled cells with unlabeled CTL at ratio 1:10 and used them in the assay. This approach allows us to follow labeled cells and to quantify % of unresponsive cells. The major factor of the unresponsiveness is the cell conditions. Other factors include monolayer imperfections and heterogeneity of the cell labeling. We incorporated this information in the Supplemental Methods Section.

We agree with the reviewer that there are variations in the amount of fluorophore picked up by individual cells during labeling. This is a major obstacle for analysis of calcium response by Flow Cytometry as we have stated in the Introduction. In this assay, the fluorescence intensity of each individual cell is compared before and after stimulation that alleviated problems with heterogeneity of the labeling. The brightest cells are presumably apoptotic cells with leaky membrane. These cells are always present in very small amount and are removed from the analysis after image subtraction procedure. We have prepared 3D plot showing fluorescence intensities of labeled T cells on the monolayer before and after stimulation with antigenic peptide at different peptide concentrations. The figure is incorporated into Figure S2 and S3 of the manuscript. The 3D image provides excellent visualization of the assay and validates the results of the quantitative analysis of 2D images.

Input cell number: We typically analyzed at least three imaging fields per sample that represent totally about 15,000-30,000 CD8 T cells. This is similar to the amount of cells that is recommended to analyze in tetramer assay. We usually observed a few false positive cells per field. The false positive signals occur mostly due to movement of non-adhered cells after peptide injection or accidental cell burst or transient rise of intracellular Ca²⁺. We also analyzed the assay linearity by diluting the antigen specific T cells and found that the linearity of the assay is preserved for tested CTL (Figure S4).

Peptide concentration: For strong agonist peptide, Ca²⁺ response has very similar characteristics within the peptide concentration range from 10⁻⁴ M to 10⁻⁷ M (Fig. 4). At the suboptimal peptide concentration the magnitude of response declined in each individual cell (Figure S3) that increases the assay error.

Time dependence: The number of false positive cells slightly increased with time mostly due to non-adherent cells. To minimize the method error, we calculate the frequency of the cells responding to strong agonist peptide at 3-4 min after the peptide injection, an optimal time.

Temperature dependence: Although a response curve has similar shape at different temperatures, higher temperature leads to increased leak of fluorophore from labeled cells and enhanced cell motility. Both factors significantly hinder image analysis, so all quantitative measurements were done at room temperature.

Comment: The revisions and explanations are fine. The 3D plots seem to be a good data representation; namely after background subtraction and using normalized scales they convey a more detail view than 2D plots, which is especially useful when comparing data sets.

Answer 4: We tested CMV-specific CTLs using ELISPOT and tetramer staining, and the results are incorporated into the revised manuscript (Table S2, Fig. S6 and Supplemental methods). ELISPOT results for healthy donor: Background reading 15{plus minus}11 (mean{plus minus}SD) per

200,000 PBMC; response to CMV specific NLVPMVATV peptide is 509.25{plus minus}17.46

(mean{plus minus}SD) per 200,000 PBMC. The frequency of NLVPMVATV-HLA-A02 tetramer positive cells was found to be 10% among total CD8+CD3+ T cells.

In agreement with previously published data (Clinical and Developmental Immunology, article ID 451059, 2012), the tetramer staining reveal larger amount of CMV-specific CD8 T cells in healthy donor as compared to ELISPOT assay. As we expected, the frequency of the CMV-specific T cells determined by CaFlux assay was higher compared to ELISPOT assay and lower than that found with tetramer staining. We incorporated these data into the revised manuscript.

Comment: These explanations and revisions answer the comments and concerns.

Answer 5: We agree and we expect that anergic and exhausted/apoptotic cells will not respond in the assay. However, our goal is to detect functional responses that are linked to the ability of T cells to exercise efficient virus- and cancer-specific responses. We propose that the efficient T-cell responses are expected to correlate with the ability of human immune system to fight viruses and cancer.

Comment: The argument is fine; it would be good to mention in this in the revised manuscript.

Minor Comments

Answer 1: We have revisited figure legends and made other necessary changes to make the figures more informative minimizing their dependence on the text.

Comment: Very good.

Answer 2: This issue has been addressed above (see answer to major question 2).

Comment: Agreed.

Answer 3: We indicated peptide concentrations in Fig. 5 Legend. Fig. 5A: 10^{-4} M of the peptide; Fig 5B: 8.75×10^{-5} M of the peptide mixture or 6×10^{-6} of each individual peptide. Since the pool contains NLVPMVATV peptide restricted by HLA-A2, we also utilized the peptide pool to induce the response in CD8 T cells from healthy donor. The frequency of the cells responding to the individual peptide and the peptide pool and the kinetics of the responses were found to be similar (Fig. S5). We plotted the average fluorescence intensity upon time and found that the kinetics of the responses remained the same (Fig. S7). However the differences in the average fluorescent intensity over background were smaller due to relatively low frequency of the responding cells.

Comment: Makes sense - agreed

Answer 4: We overlooked this paper. We quote this work in the revised manuscript.

Comment: good

Answer 5: The peptide concentrations have been indicated in the Figure Legend consistent with our answer to the Reviewer's minor comment #1. We equalized scales of x- and y-axes.

Comment: good

Reviewer #2 (Remarks to the Author):

The authors satisfactorily addressed several important issues.

The manuscript is substantially improved.

Nevertheless, a few points should be clarified.

1) The four movies uploaded by the authors in response to my request are apparently not numbered (although they can be identified by their description). The authors should please upload numbered movies.

One problem with Movie 3 and 4 is that the two movies were performed using two different reagents to stimulate T cells. Since the HCMV peptide pool from ProImmune contains the NLVPMVATV peptide, why was the ProImmune pool not used for both experiments?

Another question concerns the fact that the healthy donor and the patient express two different MHC (the healthy donor is HLA-A*02:01 positive while the patient is HLA-A*01:01 positive).

Therefore, the differences observed in the $[Ca^{2+}]_i$ kinetics might be due to the differences in peptide binding to the different MHC. A control performed using a HLA-A*01:01 healthy donor might be required.

2) On page 10 of the revised manuscript the authors suggest that the discrepancy between the responses observed using tetramer staining and the responses observed using the CaFlux or the ELISpot assays might be due to the fact that part of the NV9-HLA-A2+ positive cells are unresponsive. I disagree with this suggestion. Discrepancy between tetramer staining and ELISpot assay might be due to the fact that ELISpot measures the production of only one cytokine and not of other T cell responses. It might be possible, for instance, that a fraction of tetramer positive T cells do not produce IFN- γ but are able to elicit cytotoxicity. The fact that the CaFlux assay provides a lower frequency of specific T cells when compared to tetramer staining can be, at least in part, explained by the fact that the T cells do not form perfect monolayers (this can be clearly appreciated in the movies). As a consequence, a lower frequency of responding cells, when compared to tetramer staining, might be due to the incapacity of some isolated T cells to present the antigen to each other during the analyzed 10-15 minutes. This point should be discussed since it is relevant when comparing the CaFlux assay with other available assays.

3) On page 10 line 14 the authors write: "However, the frequency of CMVspecific T cells measured by ELISpot assay was about 2-times larger (Table S2)." This might be a typo, since the frequency measured by ELISpot assay seems to be 2-times lower.

Reviewer #3 (Remarks to the Author):

The authors have satisfactorily addressed the major concerns raised during the last round of review.

REVIEWERS' COMMENTS:

Reviewer #1 (Remarks to the Author):

Answer 1: The reviewer's comment is well taken. It is true that some CTL responses do not depend on Ca^{2+} signaling. However, 75% of all activation-regulated genes showed dependence on Ca^{2+} flux (Nature Immunol, 2:316-324, 2001). This emphasizes the role of Ca^{2+} signaling in regulating early signaling events, which influence functional CTL responses (Immunology and Cell Biology, 93:694-704, 2015). Moreover, efficiency of cytolytic activity, the major function of CTL, clearly depends on Ca^{2+} signaling. We have previously shown that the kinetics of Ca^{2+} mobilization, regardless of how strong Ca^{2+} signal might follow later, determines the kinetics of cytolytic vesicles delivery to the CTL/target cell interface, which is linked to the kinetics of target cell destruction (Immunity, 31: 632-642, 2009; Sci Signal., 3: pe50, 2010; Immunol. Res., 51: 183-94, 2011). It has been shown that recognition of a strong agonist ligand results in a very rapid onset and fast accumulation of intracellular Ca^{2+} , while stimulation with weak agonists significantly delays the onset time and induces much slower accumulation of intracellular Ca^{2+} with lower average magnitude (J. Exp. Med., 185:1815-1825, 1997; J. Immunol., 184:1829-1839, 2010). Furthermore, the delay of the onset time revealed a poor dependence on the peptide concentration (J. Exp. Med., 185:1815-1825, 1997). This is consistent with the data presented in Figure 3. Recognition of a strong agonist ligand, which induced sensitive and robust cytolytic response, resulted in a very rapid onset and fast accumulation of intracellular Ca^{2+} , while weak agonists induced much slower accumulation of intracellular Ca^{2+} that is linked to a profound decrease of cytotoxic potency. The data presented on Fig. 3 were produced at very high peptide concentration (10⁻⁴ M), i.e., conditions at which the difference in T-cell responses may be diminished. Nevertheless, the difference in the kinetics of Ca^{2+} accumulation for strong and weak agonist ligands is clearly evident. In addition, analysis of the 3D plot images at suboptimal peptide concentration (10⁻⁸ M) of cognate peptide (Fig. 4 and Supplemental Figure 2) revealed that the magnitude of the response reduced due to a lower amplitude of individual cell responses, but not due to the changes in the number of the responding cells. We clarified these issues in the revised manuscript.

Comment: The revisions are fine. It should be mentioned that CTL have two cytotoxic mechanisms; one of which, the Fas/FasL pathway is not or very little Ca²⁺ dependent and hence is not detected by the described method (PMID: 24594598; PMID: 9522460; PMID: 9529322, PMID: 8666927). This cytotoxic pathway is important, e.g. for tumor control (PMID: 23341634).

Answer 1.1: We included appropriate considerations and references into the revised manuscript to address the reviewer concern.

Answer 2: The reviewer correctly noted that the specificity of T cells for 100 different peptides couldn't be determined in one round of the assay. What we really meant is that 100 different peptides could be tested in one round of the assay, which is important for testing of peptide pools in clinical applications. Smaller groups of the peptides have to be analyzed to nail down which peptides serve as T cell epitopes for a sample of polyclonal T cells. We agree that peptides with significantly lower affinity may be missed. However, usually peptides that form sufficiently stable complexes with MHC proteins could induce potent T cell response and play essential role in the immune defense. The effect of the peptide competition on the stimulation of functional T cell response was studied with peptide pool comprising 32 well-characterized viral epitopes (Viruses, 4: 2636-2649, 2012). The effect of the competition was minor, - it causes around 20% decrease in the magnitude of the response. We have incorporated necessary comments into the revised manuscript to clarify this issue.

Comment: The arguments and revisions are fine. It is good to make clear these issues, namely that multiple rounds are needed to identify peptide specificities.

Answer 2.1: We reiterated in the revised manuscript that a mixture of peptides could be tested in one round of analysis, but peptide(s) identification requires multiple rounds of analysis.

Answer 3: Even though cloned T cells expressed the same TCR, they are not uniform. At the time when the T cells are tested after the stimulation, there are never 100% of the tested T cells capable to mount a response to cognate pMHC. For example, about 10% of unresponsive cells were observed even for Jurkat cells stimulated with surface-bound anti-CD3 antibodies (PLOS One, 9: e85934, 2014). There are various reasons for the unresponsiveness, namely, apoptosis, anergy, difference in the number of cell surface receptors etc. To measure CTL responsiveness in the CaFlux assay, we diluted the labeled cells with unlabeled CTL at ratio 1:10 and used them in the assay. This approach allows us to follow labeled cells and to quantify % of unresponsive cells. The major factor of the unresponsiveness is the cell conditions. Other factors include monolayer imperfections and heterogeneity of the cell labeling. We incorporated this information in the Supplemental Methods Section.

We agree with the reviewer that there are variations in the amount of fluorophore picked up by individual cells during labeling. This is a major obstacle for analysis of calcium response by Flow Cytometry as we have stated in the Introduction. In this assay, the fluorescence intensity of each individual cell is compared before and after stimulation that alleviated problems with heterogeneity of the labeling. The brightest cells are presumably apoptotic cells with leaky membrane. These cells are always present in very

small amount and are removed from the analysis after image subtraction procedure. We have prepared 3D plot showing fluorescence intensities of labeled T cells on the monolayer before and after stimulation with antigenic peptide at different peptide concentrations. The figure is incorporated into Figure S2 and S3 of the manuscript. The 3D image provides excellent visualization of the assay and validates the results of the quantitative analysis of 2D images.

Input cell number: We typically analyzed at least three imaging fields per sample that represent totally about 15,000-30,000 CD8 T cells. This is similar to the amount of cells that is recommended to analyze in tetramer assay. We usually observed a few false positive cells per field. The false positive signals occur mostly due to movement of non-adhered cells after peptide injection or accidental cell burst or transient rise of intracellular Ca²⁺. We also analyzed the assay linearity by diluting the antigen specific T cells and found that the linearity of the assay is preserved for tested CTL (Figure S4).

Peptide concentration: For strong agonist peptide, Ca²⁺ response has very similar characteristics within the peptide concentration range from 10⁻⁴ M to 10⁻⁷ M (Fig. 4). At the suboptimal peptide concentration the magnitude of response declined in each individual cell (Figure S3) that increases the assay error.

Time dependence: The number of false positive cells slightly increased with time mostly due to non-adherent cells. To minimize the method error, we calculate the frequency of the cells responding to strong agonist peptide at 3-4 min after the peptide injection, an optimal time.

Temperature dependence: Although a response curve has similar shape at different temperatures, higher temperature leads to increased leak of fluorophore from labeled cells and enhanced cell motility. Both factors significantly hinder image analysis, so all quantitative measurements were done at room temperature.

Comment: The revisions and explanations are fine. The 3D plots seem to be a good data representation; namely after background subtraction and using normalized scales they convey a more detail view than 2D plots, which is especially useful when comparing data sets.

Answer 3.1: *We are delighted that our answer is satisfactory.*

Answer 4: We tested CMV-specific CTLs using ELISPOT and tetramer staining, and the results are incorporated into the revised manuscript (Table S2, Fig. S6 and Supplemental methods). ELISPOT results for healthy donor: Background reading 15{plus minus}11 (mean{plus minus}SD) per

200,000 PBMC; response to CMV specific NLVPMVATV peptide is 509.25{plus minus}17.46 (mean{plus minus}SD) per 200,000 PBMC. The frequency of NLVPMVATV-HLA-A02 tetramer positive cells was found to be 10% among total CD8+CD3+ T cells.

In agreement with previously published data (Clinical and Developmental Immunology, article ID 451059, 2012), the tetramer staining reveal larger amount of CMV-specific CD8 T cells in healthy donor as compared to ELISPOT assay. As we expected, the

frequency of the CMV-specific T cells determined by CaFlux assay was higher compared to ELISPOT assay and lower than that found with tetramer staining. We incorporated these data into the revised manuscript.

Comment: These explanations and revisions answer the comments and concerns.

Answer 4.1: We are delighted that our answer is satisfactory.

Answer 5: We agree and we expect that anergic and exhausted/apoptotic cells will not respond in the assay. However, our goal is to detect functional responses that are linked to the ability of T cells to exercise efficient virus- and cancer-specific responses. We propose that the efficient T-cell responses are expected to correlate with the ability of human immune system to fight viruses and cancer.

Comment: The argument is fine; it would be good to mention in this in the revised manuscript.

Answer 5.1: We discussed this issue in more details in the revised manuscript

Minor Comments

Answer 1: We have revisited figure legends and made other necessary changes to make the figures more informative minimizing their dependence on the text.

Comment: Very good.

Answer 2: This issue has been addressed above (see answer to major question 2).

Comment: Agreed.

Answer 3: We indicated peptide concentrations in Fig. 5 Legend. Fig. 5A: 10^{-4} M of the peptide; Fig 5B: 8.75×10^{-5} M of the peptide mixture or 6×10^{-6} of each individual peptide. Since the pool contains NLVPMVATV peptide restricted by HLA-A2, we also utilized the peptide pool to induce the response in CD8 T cells from healthy donor. The frequency of the cells responding to the individual peptide and the peptide pool and the kinetics of the responses were found to be similar (Fig. S5). We plotted the average fluorescence intensity upon time and found that the kinetics of the responses remained the same (Fig. S7). However the differences in the average fluorescent intensity over background were smaller due to relatively low frequency of the responding cells.

Comment: Makes sense - agreed

Answer 4: We overlooked this paper. We quote this work in the revised manuscript.

Comment: good

Answer 5: The peptide concentrations have been indicated in the Figure Legend consistent with our answer to the Reviewer's minor comment #1. We equalized scales of x- and y-axes.

Comment: good

Answers to minor comments: *We are delighted that our answers to minor comments are satisfactory*

Reviewer #2 (Remarks to the Author):

The authors satisfactorily addressed several important issues.
The manuscript is substantially improved.

Nevertheless, a few points should be clarified.

1) The four movies uploaded by the authors in response to my request are apparently not numbered (although they can be identified by their description). The authors should please upload numbered movies.

One problem with Movie 3 and 4 is that the two movies were performed using two different reagents to stimulate T cells. Since the HCMV peptide pool from ProlImmune contains the NLVPMVATV peptide, why was the ProlImmune pool not used for both experiments?

Another question concerns the fact that the healthy donor and the patient express two different MHC (the healthy donor is HLA-A*02:01 positive while the patient is HLA-A*01:01 positive). Therefore, the differences observed in the $[Ca^{2+}]_i$ kinetics might be due to the differences in peptide binding to the different MHC. A control performed using a HLA-A*01:01 healthy donor might be required.

Answer 1.2: *We uploaded numbered movies, i.e., Movie1, Movie2 etc., during first resubmission of the manuscript. Corresponding movie legends have been provided in the Supplemental Information. We will make every effort making sure that movies upload during second resubmission are numbered.*

We have included additional movie illustrating responses of T cells from healthy donor to the ProlImmune peptide pool into the Supplemental Information of the revised manuscript. The response appears to be very similar to that induced by single HCMV-derived peptide NLVPMVATV.

*The peptides used for analysis of HCMV-specific T cell responses are immunodominant peptides that normally interact with corresponding MHC proteins with high affinity. Indeed, HLA-A*0101 and HLA-A*0201 restricted HCMV-derived peptides, which are included in HCMV-specific ProlImmune peptide pool, possess similar binding characteristics as established experimentally (J. Immunother. 2012; 35(2):142-53; Large scale analysis of peptide-HLA-I stability: <http://www.iedb.org/refId/1028282>) and from theoretical calculations (Immunology, 2014, 141(1): 18–26). We included this information into the revised manuscript.*

2) On page 10 of the revised manuscript the authors suggest that the discrepancy between the responses observed using tetramer staining and the responses observed using the CaFlux or the ELISpot assays might be due to the fact that part of the NV9-HLA-A2+ positive cells are unresponsive. I disagree with this suggestion. Discrepancy between tetramer staining and ELISpot assay might be due to the fact that ELISpot measures the production of only one cytokine and not of other T cell responses. It might be possible, for instance, that a fraction of tetramer positive T cells do not produce IFN- γ

but are able to elicit cytotoxicity. The fact that the CaFlux assay provides a lower frequency of specific T cells when compared to tetramer staining can be, at least in part, explained by the fact that the T cells do not form perfect monolayers (this can be clearly appreciated in the movies). As a consequence, a lower frequency of responding cells, when compared to tetramer staining, might be due to the incapacity of some isolated T cells to present the antigen to each other during the analyzed 10-15 minutes. This point should be discussed since it is relevant when comparing the CaFlux assay with other available assays.

Answer 2.2: We agree with the reviewer that “Discrepancy between tetramer staining and ELISpot assay might be due to the fact that ELISpot measures the production of only one cytokine and not of other T cell responses”. Indeed we stated in the manuscript that “the ELISpot assay counts only INF-g producing cells, while the CaFlux assay detects all responding cells independently of their functionality”. We reiterated this point in the revised manuscript.

We also agree with the reviewer that imperfection of the monolayer might influence the ability of some T cells to recognize an antigen. However, as evident from the comparison of the bright field and fluorescent images of CER43 cells (see Supplementary Methods), most of non-responding cells, which constitute less than 10% of the population, are apoptotic or dying cells, and much smaller fraction of the non-responders caused by imperfection of the monolayer. Thus, the imperfection of a monolayer is unlikely responsible for two times difference between the number of tetramer positive T cells and number of responding cells in CaFlux assay in our experiments. We would like to suggest that the observed difference is due to presence of different kind of unresponsive cells such as apoptotic cells, terminally differentiated exhausted cells, cells with low level of TCR and/or co-receptors etc. We will incorporate these considerations into the revised manuscript. To address this issue comprehensively, careful comparison studies should be performed, which is beyond the scope of this manuscript. Future application of CaFlux and ELISpot assays to the analysis of many more samples of human T cells will allow us to answer this question more definitively.

3) On page 10 line 14 the authors write: "However, the frequency of CMVspecific T cells measured by ELISpot assay was about 2-times larger (Table S2)." This might be a typo, since the frequency measured by ELISpot assay seems to be 2-times lower.

Answer 3.2: We corrected the typo.

Reviewer #3 (Remarks to the Author):

The authors have satisfactorily addressed the major concerns raised during the last round of review.

Answer to reviewer’s comments: We are delighted that our answers to initial critique are satisfactory